# Seaweed Supplementation Failed to Affect Fecal Microbiota and Metabolome as Well as Fecal IgA and Apparent Nutrient Digestibility in Adult Dogs

**DOI:** 10.3390/ani11082234

**Published:** 2021-07-29

**Authors:** Carlo Pinna, Carla Giuditta Vecchiato, Monica Grandi, Claudio Stefanelli, Augusta Zannoni, Giacomo Biagi

**Affiliations:** 1Department of Veterinary Medical Sciences, University of Bologna, via Tolara di Sopra 50, 40064 Ozzano Emilia, Italy; carlo.pinna2@unibo.it (C.P.); carla.vecchiato2@unibo.it (C.G.V.); augusta.zannoni@unibo.it (A.Z.); giacomo.biagi@unibo.it (G.B.); 2Department for Life Quality Studies, University of Bologna, Corso d’Augusto 237, 47921 Rimini, Italy; claudio.stefanelli@unibo.it

**Keywords:** marine macroalgae, prebiotics, immunomodulation

## Abstract

**Simple Summary:**

Seaweeds represent a source of bioactive compounds that have recently drawn the attention of the scientific community for their possible application as health-promoting ingredients. In particular, their indigestible polysaccharides have exhibited promising prebiotic properties both in humans and in farm animals. The present study investigated for the first time in healthy adult dogs the effects of dietary supplementations with intact seaweeds (brown algae *Ascophyllum nodosum*, *Undaria pinnatifida*, *Saccharina japonica* and red alga *Palmaria palmata*) on some fecal bacterial populations and metabolites, fecal IgA and apparent total tract digestibility of nutrients. The different algal supplementations (fed to ten healthy adult dogs at a dietary dose of 15 g/kg for 28 days) did not have any significant effect on the selected fecal microbial parameters, intestinal immunity and nutrient digestibility. Further studies investigating higher dietary inclusions of intact seaweeds or their polysaccharide extracts are needed to gain a better understanding of the potential of these interesting marine resources in companion animal nutrition.

**Abstract:**

The present study investigated in dogs the dietary effects of intact seaweeds on some fecal bacterial populations and metabolites, fecal IgA and apparent total tract digestibility (ATTD). Ten healthy adult dogs were enrolled in a 5 × 5 replicated Latin square design to evaluate five dietary treatments: control diet (CD); CD + *Ascophyllum nodosum*; CD + *Undaria pinnatifida*; CD + *Saccharina japonica*; CD + *Palmaria palmata* (*n* replicates per treatment = 10). Seaweeds were added to food at a daily dose of 15 g/kg. The CD contained silica as a digestion marker. Each feeding period lasted 28 d, with a 7 d wash-out in between. Feces were collected at days 21 and 28 of each period for chemical and microbiological analyses. Fecal samples were collected during the last five days of each period for ATTD assessment. Dogs showed good health conditions throughout the study. The fecal chemical parameters, fecal IgA and nutrient ATTD were not influenced by algal supplementation. Similarly, microbiological analyses did not reveal any effect by seaweed ingestion. In conclusion, algal supplementation at a dose of 15 g/kg of diet failed to exert noticeable effects on the canine fecal parameters evaluated in the present study.

## 1. Introduction

The crucial role that the intestinal microbiota plays in supporting host health is a well-established concept [1] and in recent decades scientific research has widely investigated different nutritional strategies aimed to positively influence the microbial ecosystem of the gastrointestinal tract [2]. Among the several dietary components investigated in this context, in particular, undigestible carbohydrates (i.e., oligosaccharides) have been explored both in humans [3,4] and other animal species such as dogs [5,6], mostly for their potential benefits on the composition and activity of the gut microbiota, leading the way to the nutritional “prebiotic” approach aimed to maintain or improve host health [7].

In this context, marine macroalgae (seaweeds), plant-like multi-cellular organisms representing a substantial part of the ocean biomass and a traditional food for some Asian human populations, have recently drawn the attention of the scientific community for their possible application as health-promoting ingredients both in human diets [8] and animal feeds [9]. Seaweeds, in fact, represent an interesting source of undigestible polysaccharides, bioactive compounds (including polyphenols, fatty acids and carotenoids) and, in some cases, proteins. Moreover, due to this last characteristic, they have been proposed as encouraging sustainable alternatives to conventional animal feed resources, particularly in the aquaculture sector [10].

Macroalgae comprise thousands of species that are classified as brown (*Phaeophyta*), red (*Rhodophyta*) and green (*Chlorophyta*) algae according to their pigmentation. Their major components are minerals [11] and a fiber fraction [12]. The latter mainly consists of undigestible polysaccharides (e.g., alginate, laminarins and fucoidans in brown algae; carrageenans and agar in red algae; ulvans in green algae) that have recently attracted scientific interest due to their biological properties such as antitumor, antiadhesive, antioxidant, antitoxin, immunomodulatory and anti-infection effects [13]. Moreover, these last components, providing resistance to gastric acidity and host digestive processes, appear promising as prebiotic substrates [14]. In this regard, several in vitro and in vivo studies investigating the prebiotic potential of intact seaweeds or their polysaccharide extracts have been published in recent years [15,16,17].

In monogastric farm animals like pigs, broilers and fish, some interesting positive effects on intestinal microbiota and the local immune system, in particular of intact brown seaweeds and their extracts, have been described [10,18,19]. Conversely, concerning companion animals such as dogs and cats, studies are rather limited and mainly focused on the evaluation of the effect of some algal polysaccharides commonly used as gelling agents in wet pet food, such as alginate [20] and carrageenan [21], on apparent nutrient digestibility. Furthermore, some studies investigating the efficacy of *Ascophyllum nodosum* in maintaining oral health [22,23] and in preserving palatability [24] have been published in recent years. On the other hand, to the authors’ knowledge, studies evaluating the effects of seaweeds on the intestinal environment in dogs and cats are have not be carried out.

The aim of the present study was to evaluate the effects deriving from dietary supplementations with different intact brown seaweeds (*Ascophyllum nodosum* (*AN*), *Undaria pinnatifida* (UP) and *Saccharina japonica* (*SJ*)) or one intact red seaweed (*Palmaria palmata* (*PP*)) on some fecal microbial populations and metabolites, fecal IgA and apparent total tract digestibility (ATTD) of nutrients in healthy adult dogs. We supposed that these intestinal functionality-associated parameters would be positively influenced by algal supplementation.

## 2. Materials and Methods

### 2.1. Animals and Diets

Ten healthy household adult dogs were involved in the study. They were between 1 and 6 years of age, of different breeds (five small to medium-sized cross-breed dogs, three Border Collies, one Australian Shepherd and one Labrador Retriever). Concerning gender and reproductive status, there were eight spayed females, one castrated male and one intact male. Their average body weight ± standard deviation was 22.8 ± 6.1 kg. Their median body condition score was 4.5 (range from 4 to 5), according to the 9-point scale proposed by Laflamme et al. [25]. During the trial, dogs remained in their usual environment. All the dogs were regularly vaccinated and periodically treated for intestinal parasites; dogs did not display gastrointestinal signs during the previous year. Owners were asked to monitor the feeding behavior, the maintenance of body weight and the stool appearance and consistency of their dog and to report any change occurring during the study.

One commercial extruded and complete diet formulated for adult dogs (Effeffe Pet Food S.p.A., Padua, Italy), containing dehydrated poultry protein, rice, maize, wheat, animal fats, fish meal, linseed, sunflower oil, dehydrated eggs, sodium, phosphate, potassium chloride and sodium chloride, was used as the control diet (CD) for the study. Four different experimental diets were set up, through the supplementation of the CD with four intact seaweeds products. In particular, it was decided to use three brown seaweeds: AN, UP and SJ, and one red seaweed: PP. In order to remove any contaminants, the algae were washed with tap water and rinsed twice with deionized water. Finally, algae were freeze-dried and ground through a 1 mm screen. The algal supplements were added at a concentration of 15 g/kg of as-is diet. Since the seaweeds were in form of a powder, they were spread over the kibble together with a small amount of water to ensure their adhesion and total consumption. Dogs were fed twice a day. Consequently, the daily dose of supplements was divided and added equally to each of the two planned daily meals. No snack or other food was offered to the dogs during the whole study period. The CD did not include significant amounts of soluble fiber sources. Silica was included at a dose of 5 g/kg as a source of acid-insoluble ash to be used as digestion marker. The dogs’ daily ration was set up by considering the energy content of the CD (determined through the modified Atwater conversion factors) and the daily energy requirements, according to the recommendations for the maintenance of adult dogs with a moderate activity level: 120 kcal/kg BW^0.75^ [26]. Prior to the beginning of the study, the determination of critical trace elements and potentially dangerous metals was performed to check their concentration in the four intact seaweeds. The chemical analysis of the CD and the algal supplements is reported in Table 1.

### 2.2. Experimental Design and Sample Collection

The feeding trial was planned as a 5 × 5 replicated Latin square design. At the beginning of the study, all dogs were adapted to the basal diet for 30 days. During the crossover study, groups of two dogs received each of the five dietary treatments (CD and the four seaweed-supplemented diets). Each feeding period lasted 28 days, with a 7-day wash-out in between periods during which all dogs received the CD. On days 21 and 28 of each feeding period, a fresh fecal sample was collected within 30 min of defecation and then frozen at −80 °C for chemical (dry matter (DM), ammonia, short-chain fatty acids (SCFAs), branched-chain fatty acids (BCFAs) and biogenic amines) and microbiological analyses. A sub-sample of feces collected at both time-points (day 21 and 28) was freeze-dried for IgA measurement. During the last five days of each feeding period (from day 24 to 28), fecal samples were pooled and frozen at −20 °C and successively freeze-dried for nutrient analyses and apparent total tract digestibility assessment.

### 2.3. Chemical Analyses and ATTD Assessment

The nutrients in the CD, intact seaweeds and freeze-dried fecal samples collected during the last 5 days of each feeding period were quantified according to AOAC International standard methods [27] (method 954.01 for crude protein, method 942.05 for crude ash, method 965.17 for phosphorus; only in CD and seaweeds: method 950.46 for water, method 920.39 for ether extract, method 991.43 for total, soluble and insoluble fiber and method 962.09 for crude fiber). Acid-insoluble ash was determined according to Vogtmann et al. [28]. For the determination of macrominerals Ca, Na, Mg and K and trace elements Zn, Mn, Fe and Cu, ash samples of the CD, algal supplements and freeze-dried feces were analyzed by atomic absorption spectrometry according to an International Standards method (EN ISO 6869:2000) [29]. Metal and other trace element (Se, As and Sb) assessment in seaweeds was performed through inductively coupled plasma–mass spectrometry (ICP-MS), following the method described by Carpenè et al. [30].

Apparent total tract DM digestibility of the experimental diets was measured through the following equation:(1)100−[(100×% marker in the diet)/% marker in feces]

Apparent total tract digestibility of each nutrient was established through the following equation:(2)100−[% nutrient in feces×(100−% DM digestibility)/% nutrient in the diet]

Since the seaweed-based supplements contained a certain amount of nutrients (ash and silica, in particular, and proteins), in order to accurately calculate the ATTD coefficients, the nutrient percentage in the diet was calculated by considering both the contribution of the CD and the seaweed supplements.

Fecal pH was evaluated with a SevenMulti pH meter (Mettler Toledo, Milano, Italy) in diluted fecal samples (1:10 *w*/*v* in distilled water). Ammonia was quantified through a commercial kit (Urea/BUN—Color; BioSystems S.A., Barcelona, Spain). SCFAs and BCFAs were determined using a 2 m glass column (inner diameter, 3 mm) of 10% SP-1000 + 1% H_3_PO_4_ on 100/120 Chromosorb W AW with nitrogen as the carrier. The chromatograph was a Fisons HRGC MEGA 2 series 8560 with a flame ionization detector. Both the injector and detector temperatures were 200 °C, and the oven was 155 °C. 2-ethylbutyric acid was used as the internal standard. For the measurement of biogenic amines, feces were diluted 1:5 *w*/*v* with perchloric acid (0.3 M); biogenic amines were later separated by HPLC and enumerated through fluorimetry, as described by Stefanelli et al. [31].

### 2.4. Microbial Analyses

Bacterial genomic DNA was extracted and isolated from fecal samples (~200 mg) using the Norgen Stool DNA Isolation Kit (Norgen Biotek Corp., Thorold, ON, Canada). Isolated DNA concentration (ng/μL) and purity were measured using a DeNovix DS-11 spectrophotometer (DeNovix Inc., Wilmington, DE, USA). Template DNA was diluted to 50 ng/μL and stored at −20 °C until further analysis. Total bacteria [32], *Firmicutes* [33], *Bifidobacterium* spp., *Enterococcus* spp. [34], *Lactobacillus* spp. [35], *Clostridium* cluster I [36], *Escherichia coli* [37] and *Faecalibacterium prausnitzii* [38], were quantified via quantitative polymerase chain reaction (qPCR) using specific primers (Table 2).

The qPCR assay was performed using a CFX96 Touch thermal cycler (Bio-Rad, Hercules, CA, USA).

Amplification was performed in duplicate for each bacterial group within each sample, while standard curves were run in triplicate.

Briefly, the PCR reaction contained 7.5 μL 2× SensiFAST No-ROX PCR MasterMix (Bioline GmbH, Luckenwalde, Germany), 4.8 μL of nuclease-free water, 0.6 μL of each 10 pmol primer and 1.5 μL of template DNA for a final reaction volume of 15 μL. The amplification cycle was as follows: initial denaturation at 95 °C for 3 min, 95 °C for 5 s, primer annealing at 55–64 °C for 10 s and 72 °C for 8 s. The cycle was repeated 40 times. A negative control (without the DNA template) was also run for each primer pair. Standard curves were constructed from eight tenfold dilutions for total bacteria, *Firmicutes*, *bifidobacteria*, *enterococci*, *lactobacilli*, *Clostridium* cluster I, *Escherichia coli* and *Faecalibacterium prausnitzii*. Cycle threshold values were plotted against standard curves for the quantification of the target bacterial DNA. Melting curves were checked after amplification to ensure the single product amplification of a consistent melting temperature.

### 2.5. Fecal IgA Determination

The determination of IgA content in lyophilized fecal samples was carried out by using a commercial kit (dog IgA ELISA Quantitation Set, Bethyl Laboratories Inc., Montgomery, TX, USA; assay range: 15.6–1000 ng/mL), following the procedure described by Zannoni et al. [39].

### 2.6. Statistical Analyses

Firstly, for each parameter, data measured after days 21 and 28 of each feeding period were compared by a Student’s *t*-test. Since no significant difference was observed, mean values obtained from each dog at 21 and 28 d were considered for statistical analysis. Data were assessed for normality through a D’Agostino and Pearson omnibus normality test. When they were not normally distributed, a logarithmic transformation was applied to normalize the data distribution. Normalized data were analyzed by a general linear model with diet as a fixed effect and animal and period as random effects. The Dunnett test was used as the post-test. When microbiological analysis revealed unquantifiable outcomes, “zero” values were assigned.

Levels of significance and tendency were set at *p* ≤ 0.05 and 0.05 < *p* < 0.1, respectively. Statistical analyses were performed using Statistica 10.0 software (Stat Soft Italia, Padua, Italy).

## 3. Results

Dog owners did not report any modification in body weight or fecal appearance and consistency of their animals during the study. All the dogs maintained good health conditions throughout the trial and entirely consumed the assigned daily ration and the algal supplements.

The chemical parameters evaluated in fecal samples did not show any influence by the dietary supplementation with intact seaweeds (Table 3).

The microbial populations evaluated in the present study were not significantly affected by treatments (Table 4). Concerning ATTD coefficients, seaweeds did not influence the digestibility of the macronutrients and most of the minerals evaluated. Only Ca digestibility showed a tendency to be influenced by algal supplementation (*p* = 0.07). In particular, compared with the CD, ATTD of Ca was improved when dogs received the AN diet (61.8% vs. 40.1%; *p* = 0.03) (Table 5).

Moreover, the concentration of fecal IgA did not significantly change among dietary treatments (Figure 1).

## 4. Discussion

The purpose of this study was to investigate the effect of dietary supplementations with intact seaweed-based products on the gastrointestinal environment of adult dogs through the determination of ATTD, fecal IgA and selected fecal bacterial populations and metabolites.

In general, the results revealed a negligible influence exerted by the selected seaweeds on canine fecal microbiota. In particular, the lack of significant effects on bacterial metabolites such as SCFAs, BCFAs and biogenic amines reflected the absence of modulation in the microbial populations evaluated in fecal samples.

Differently from the present study, several literature reviews, describing the results of both in vitro and in vivo studies, have recently emphasized interesting prebiotic effects deriving from dietary supplementation with intact seaweeds (such as *Saccharina*, *Laminaria*, *Ascophyllum* and *Undaria* spp.) or their major undigestible polysaccharide components (namely, alginate, fucoidan and laminarin extracted from brown algae) [12,16,17,18].

Recently, a growing interest in the potential application of marine macroalgae or their extracts in farm animal nutrition has been observed. In pigs, for example, brown algae are considered promising dietary supplements (and safe alternatives to antibiotic growth promoters), able to exert prebiotic and immunomodulatory effects on the intestinal environment [19,40]. In fact, differently from the present study, previous investigations in weaning or weaned piglets fed seaweed extracts such as laminarins (0.3 g/kg) and fucoidans (0.24 g/kg) from the brown algae AN and *Laminaria* spp. have shown positive changes in the intestinal microbiota, such as increased *Lactobacillus* and *Bifidobacterium* spp., decreased *Enterobacteria*, higher concentration of SCFAs, acidification of the pH and/or a reduction in ammonia in the hindgut [10]. Moreover, laminarin-rich extracts deriving from AN and *Laminaria hyperborean* have shown interesting antimicrobial activity against *E. coli*, *Salmonella typhimurium*, *Staphylococcus aureus* and *Listeria monocytogenes* [41]. Likewise, a study by Patra et al. [42] described interesting antibacterial properties exerted by an essential oil extracted from UP on pathogenic *Salmonella*
*typhimurium* and *Staphylcoccus aureus* bacterial strains.

Additionally, concerning red seaweeds, both as whole biomass or polysaccharide extracts (i.e., carrageenans and agarans), interesting prebiotic effects on gut microbiota have been reported [43,44]. In vitro, dietary fibers extracted from PP (mainly constituted by xylans) have been shown to stimulate microbial fermentation, enhancing SCFA production [45]. In broiler chickens, the consumption of PP at dietary doses of 18, 24 and 30 g/kg positively influenced the presence of beneficial intestinal bacteria (in particular, lactobacilli and bifidobacteria) and decreased the number of undesirable microbes like *Cl. perfringens* [46]. Similarly, feed supplementation with other red seaweeds, *Condrus crispus* and *Sarcodiotheca gaudichaudii* (at doses of 5–10 and 20 g/kg of diet), positively affected the intestinal environment of layers hens, as higher concentrations of SCFAs and a higher abundance of *Bifidobacterium longum* and *Streptococcus salivarius* together with a reduction in *C. perfringens* was observed [47]. Nevertheless, other studies investigating the effects of polysaccharide extract [48] or intact seaweeds such as *A. nodosum* at dietary doses up to 10 g/kg [49] in pigs failed to demonstrate clear effects on intestinal microbiota, similarly to the present trial.

As previously pointed out by other authors, the inconsistent results from studies on monogastric farm animals receiving seaweeds as “whole products” could be attributable to the different chemical composition of this type of supplements. In particular, their polysaccharide content can greatly differ in relation to algal species and environmental factors (not only season and temperature, but also harvest time and geographical location) [10,50]. In brown algae, the total polysaccharide content has been estimated within a wide range between 10% and 75% and in red algae between 10% and 59% [15]. The nutritional content of the algal supplements used in the present study (including the concentration of minerals and fibrous fractions) results in agreement with the values reported in the literature [10,19]. In this regard, the absence of effects observed in the dogs is difficult to explain. It could be hypothesized that, considering the absence of significant potential prebiotic ingredients in the CD, the amount of undigestible polysaccharides derived from seaweed supplementation reaching the canine hindgut was too small to exert evident modulation on the fecal parameters investigated here in this species.

Moreover, it must be underlined that in the present investigation only few of the main key taxa of the canine microbiome have been evaluated. This might represent a limitation for the study since possible changes in other bacteria could not have been detected.

In general, for ethical reasons, according to the current European legislation, feces represent the only analyzable type of sample in studies investigating the gastrointestinal tract of companion animals. Actually, fecal composition is not fully representative of the gut environment [51], even though this material necessarily contains bacteria and compounds derived from the intestinal tract. In particular, microbial fermentation occurring in the ileum of dogs may not affect colonic and fecal microbiota [52]. This aspect could at least partially explain the lack of effects from algal supplementation observed in the present study. Furthermore, the differences in terms of age, body size and living environment between the dogs might also have contributed to the outcome of this investigation. On the other hand, prebiotic effects from dietary algal substrates have been mostly described by studies carried out with more standardized and conventional “experimental” animals such as pigs and chickens [10,19], where digesta sampling from the ileum or colon is allowed by current legislation.

In this study, fecal IgA analysis revealed no effect in dogs when they received the supplemented diets. Secretory IgA is the most important humoral immune factor in the intestinal tract and represents the expression of the mucosal immune response [53], limiting, in particular, the access of commensal and potentially pathogenic bacteria to the intestinal epithelial surface and neutralizing bacterial toxins [54]. Intestinal IgA has been described to be involved in gut homeostasis maintenance [55] and gut microbiota have been shown to influence its production [56,57]. Consequently, higher fecal IgA after a prebiotic supplementation may be interpreted as an enhancement effect of beneficial bacteria and their products on the intestinal immune system [58]. Accordingly, previous studies investigating prebiotics [59] and probiotics [60] in dogs showed an improvement of intestinal IgA excretion, although the literature offers conflicting results in this regard [61,62,63]. In the present study, an increase in fecal IgA would have been explained as an expression of immune stimulatory activity exerted by seaweed supplementation, similar to previous studies in other animal species, albeit other immunological parameters such as tumor necrosis factor or interleukins have been investigated more often in this context [18,43]. Interestingly, a study by Karimi [46] highlighted that the inclusion of intact PP in broilers’ diets (at a dose of 18 g/kg) improved the secretion of IgA together with the best growth performance and taller villus height.

It is important to emphasize that algal polysaccharides (glycans) have differences in their structure in comparison to homologous terrestrial compounds [14]. Even though the catabolism of these complex molecules might potentially favor a prebiotic effect in the gut environment, it has been pointed out that the intestinal microbiota must necessarily express specific enzymes to use these marine substrates as a carbon source for bacterial fermentation [64]. Interestingly, in humans, the ability to express this particular enzymatic activity has been demonstrated to be a consequence of horizontal gene transfer from the marine environment, in particular, through the frequent consumption of seaweeds in the diet [65]. In this regard, scientific research has described important geographical differences in terms of the presence and activity of algal polysaccharide-degrading microbial enzymes. Consequently, not all human gut microbiomes have the same competencies in this regard [8] and in terrestrial domestic animals such as dogs and cats, no information is available.

The algal supplementations investigated in the present study did not influence nutrient ATTD. Only Ca digestibility displayed a tendency to be influenced by seaweeds (without reaching statistical significance), with an improvement of this last parameter when dogs received the AN diet.

Better nutrient digestibility has been associated with a beneficial influence on the gut environment from seaweed consumption. In fact, a positive modulation of intestinal microbiota exerted by these marine substrates, favoring an improvement of the production of trophic metabolites such as SCFAs (*n*-butyrate, in primis) could promote a better absorptive capacity of the intestinal epithelium [66].

Previous studies evaluating the effects of algal supplementations on ATTD in pigs have reported conflicting evidence [18]. Similar to our results, the ingestion of dried intact AN supplements failed to show an influence on diet digestibility both in weaned piglets [67] and adult pigs [68] (dietary doses of 10–20 and 2.5 g/kg, respectively). Differently, several trials have described an improvement of apparent digestibility of nutrients such as proteins, fat, minerals, neutral detergent fiber and organic and dry matter in piglets receiving dietary supplementations with variable doses of brown seaweed extracts, mainly laminarins (0.11–0.3 g/kg), fucoidans (0.09–0.24 g/kg) and their combination [19].

Presumably, the lack of effect on ATTD shown by the present trial could be partially attributable to the relatively low dietary dose of intact seaweeds provided to dogs, as the majority of their daily diet consisted of the same control basal diet (CD) during all five experimental periods. Nevertheless, it must be emphasized that the lack of influence on ATTD could be considered positively since the algal supplementations tested in the present study did not affect the relatively high digestibility of the basal diet.

## 5. Conclusions

The present study showed that the ingestion of intact seaweeds at a dose of 15 g/kg of diet was well tolerated by the dogs and it did not alter nutrient ATTD. On the other hand, it failed to exhibit noticeable effects on fecal microbiota and immunological parameters such as fecal IgA.

Despite the scientific evidence supporting a promising role for seaweeds and their extracts as “health-promoting ingredients” in farm animals, there is a limited understanding of the effective ability of canine gut microbiota to ferment algal polysaccharides since, to the authors’ knowledge, this research represents the first investigation on the effects of algal supplements on the intestinal ecosystem of the dog.

Since it has been pointed out that intestinal microbiota must necessarily express specific enzymes to use algal polysaccharides as a carbon source for bacterial fermentation [64] and only a regular seaweed consumption could promote a selective pressure on the intestinal microbiota, favoring algal polysaccharide-fermenting bacteria [69], in companion animals such as dogs, for which dietary supplementation with marine vegetables is unusual, the usefulness of seaweed-based ingredients might appear questionable. Moreover, the ability of seaweed to concentrate heavy metals from seawater potentially represents a serious constraint to their consumption. The modest results observed during the present study might suggest the inability of canine intestinal bacteria to utilize the polysaccharides contained in algal supplements, even though it must be stressed that the microbial analysis applied here did not provide an in-depth investigation of the canine fecal microbiome. Certainly, more comprehensive studies are required to investigate the utilization of higher doses of intact seaweeds or their undigestible polysaccharide extracts in pet nutrition.

## Figures and Tables

**Figure 1 animals-11-02234-f001:**
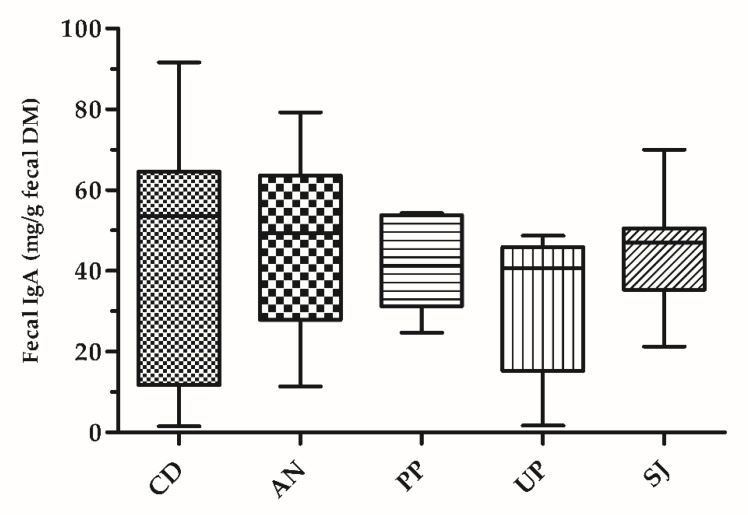
Fecal IgA concentrations (mg/g fecal DM) in dogs fed with a control diet supplemented or not with different intact seaweeds. Values are the means of 10 dogs per treatment. CD, control diet; AN, control diet supplemented with *Ascophyllum nodosum*; PP, control diet supplemented with *Palmaria palmata*; UP, control diet supplemented with *Undaria pinnatifida*; SJ, control diet supplemented with *Saccharina japonica*.

**Table 1 animals-11-02234-t001:** Chemical composition of the commercial extruded control diet (CD) and intact seaweeds used as supplements (15 g/kg of as-is diet) during the present study.

	CD	*Ascophyllum Nodosum*	*Palmaria Palmata*	*Undaria Pinnatifida*	*Saccharina Japonica*
Dry matter (g/kg)	939	910	876	795	831
On a dry matter basis (g/kg)					
Crude protein	271	51.4	142	126	117
Ether extract	164	33.5	0.77	2.65	2.83
Crude fiber	16.2	34.6	25.6	42.5	74.7
Insoluble dietary fiber	59.2	392.7	180.4	275.1	435.5
Soluble dietary fiber	18.2	156.4	167.5	53.8	115.7
Total dietary fiber	77.4	549.1	347.9	328.9	551.2
Crude ash	70.2	212	291	385	217
Acid-insoluble ash	5.37	2.19	1.09	0.49	0.30
Macrominerals (g/kg)					
*Ca*	0.175	0.215	0.058	0.104	0.169
*P*	0.085	0.005	0.019	0.032	0.010
*Mg*	0.011	0.080	0.039	0.115	0.088
*Na*	0.050	0.384	0.292	0.760	0.278
*K*	0.030	0.088	0.470	0.386	0.224
Trace minerals (mg/kg)					
*Zn*	166.3	46.3	18.1	25.7	57.5
*Mn*	42.6	12.3	58.6	5.77	4.68
*Fe*	158.5	161.1	249.9	133.7	54.4
*Cu*	12.9	n.q.	n.q.	n.q.	n.q.
Metals and other trace elements (mg/kg)					
*Pb*		0.078	1.98	0.47	0.23
*Cd*		0.30	0.05	0.22	0.20
*Cr*		0.61	1.16	0.24	0.29
*Hg*		0.02	0.02	0.02	0.03
*As*		33.8	3.84	36.1	24.2
*Al*		27.0	322	33.1	23.8
*Co*		0.43	0.30	0.18	0.08
*Ni*		1.08	1.96	2.03	0.21
*Se*		0.06	0.18	0.21	0.08
*Mo*		0.84	0.64	0.29	0.17
*Ag*		0.05	0.20	0.01	0.01
*Ti*		n.q.	0.01	n.q.	n.q.
*U*		0.62	0.10	0.51	0.45
*Sb*		0.08	0.03	0.03	0.03
*V*		1.25	12.7	0.41	2.43

n.q.: not quantified.

**Table 2 animals-11-02234-t002:** Primers used for quantitative PCR analysis.

Target Bacterial Populations	Primers	Sequence (5′–3′)	Reference
Total bacteria	UniF	CCTACGGGAGGCAGCAG	[32]
	UniR	ATTACCGCGGCTGCTGG	
Firmicutes	Firm350f	GGCAGCAGTRGGGAATCTTC	[33]
	Firm814r	ACACYTAGYACTCATCGTTT	
*Bifidobacterium* spp.	Bif_F	TCGCGTCYGGTGTGAAAG	[34]
	Bif_R	CCACATCCAGCRTCCAC	
*Enterococcus* spp.	Ent_F	CCCTTATTGTTAGTTGCCATCATT	[34]
	Ent_R	ACTCGTTGTACTTCCCATTGT	
*Lactobacillus* spp.	Lac_F	AGCAGTAGGGAATCTTCCA	[35]
	Lac_R	CACCGCTACACATGGAG	
*Clostridium* cluster I	CloI-F	TACCHRAGGAGGAAGCCAC	[36]
	CloI-R	GTTCTTCCTAATCTCTACGCAT	
*Escherichia coli*	Coli_F	GTTAATACCTTTGCTCATTGA	[37]
	Coli_R	ACCAGGGTATCTAATCCTGTT	
*Faecalibacterium prausnitzii*	Fprau 07	CCATGAATTGCCTTCAAAACTGTT	[38]
	Fprau 02	GAGCCTCAGCGTCAGTTGGT	

**Table 3 animals-11-02234-t003:** Fecal dry matter (DM) percentage and concentrations of fecal ammonia, SCFAs, BCFAs (μmol/g fecal DM) and biogenic amines (nmol/g fecal DM) in dogs fed with a control diet supplemented or not with different intact seaweeds.

Chemical Parameters	CD	AN	PP	UP	SJ	*p*-Value	Pooled SEM
DM	40.6	39.6	40.2	41.2	40.1	0.984	1.5
Ammonia	71.0	71.7	76.6	72.7	79.6	0.887	6.5
Acetic acid	125	131	144	129	149	0.562	11.6
Propionic acid °	85.9	99.3	88.1	78.4	91.3	0.738	9.5
*n*-Butyric acid °	20.7	25.1	24.6	25.1	30.5	0.271	2.9
Valeric acid °	0.47	0.16	1.21	0.67	0.45	0.414	0.36
iso-Butyric acid °	4.21	4.27	4.62	4.66	4.75	0.958	0.45
iso-Valeric acid °	5.70	6.33	6.22	6.41	6.54	0.968	0.69
Total SCFA	231	255	257	233	271	0.676	21.6
Total BCFA °	9.91	10.6	10.8	11.1	11.3	0.974	1.13
SCFA + BCFA	242	266	269	244	282	0.668	21.8
Putrescine °	3155	4398	2751	3269	3554	0.634	736
Cadaverine °	1215	1423	1233	1772	1481	0.971	454
Spermidine °	1249	1049	1206	1221	1256	0.832	136
Spermine	700	613	653	592	705	0.840	79.8

Values are the means of 10 dogs per treatment. CD, control diet; AN, control diet supplemented with *Ascophyllum nodosum*; PP, control diet supplemented with *Palmaria palmata*; UP, control diet supplemented with *Undaria pinnatifida*; SJ, control diet supplemented with *Saccharina japonica*. ° GLM analysis performed after logarithmic transformation to normalize data.

**Table 4 animals-11-02234-t004:** Microbial analysis (log copies DNA/ng DNA) of feces from dogs fed with a control diet supplemented or not with different intact seaweeds.

Bacterial Populations Item	CD	AN	PP	UP	SJ	*p*-Value	Pooled SEM
Total bacteria	5.92	5.91	6.16	6.29	6.18	0.614	0.17
*Firmicutes*	2.87	2.62	2.79	2.68	3.00	0.807	0.24
*Bifidobacterium* spp.	0.30	0.26	0.19	0.44	0.46	0.852	0.18
*Lactobacillus* spp.	n.q.	n.q.	0.08	0.02	0.20	0.190	0.07
*Faecalibacterium prausnitzii*	1.20	0.75	0.96	0.65	0.98	0.178	0.16
*Enterococcus* spp.	4.01	3.91	4.13	4.60	4.44	0.308	0.23
*Clostridium* cluster I	2.49	2.72	2.85	2.79	2.67	0.587	0.14
*E. coli*	1.38	1.09	1.15	0.95	0.99	0.898	0.34

Values are the means of 10 dogs per treatment. CD, control diet; AN, control diet supplemented with *Ascophyllum nodosum*; PP, control diet supplemented with *Palmaria palmata*; UP, control diet supplemented with *Undaria pinnatifida*; SJ, control diet supplemented with *Saccharina japonica*; n.q., not quantified.

**Table 5 animals-11-02234-t005:** Apparent total tract digestibility coefficients in dogs fed with a control diet supplemented or not with different intact seaweeds.

Nutrients Item	CD	AN	PP	UP	SJ	*p*-Value	Pooled SEM
Dry matter	87.7	90.2	89.9	90.5	89.7	0.568	1.2
Crude protein	88.6	91.2	91.0	91.2	90.7	0.523	1.2
Crude ash	50.1	65.2	61.5	63.9	60.5	0.218	4.6
*Macrominerals*							
Ca	40.1	61.8 *	58.3	50.3	57.1	**0.073**	5.32
P	39.5	52.5	55.8	50.0	51.7	0.512	6.36
Mg °	41.0	42.6	30.0	31.7	38.8	0.916	10.5
Na	96.4	97.3	98.1	97.9	97.4	0.108	0.41
K	95.7	95.4	94.8	94.7	95.5	0.749	0.63
*Trace minerals*							
Zn	16.7	39.6	35.4	32.8	35.6	0.406	8.42
Mn	12.8	34.0	33.9	25.2	30.1	0.420	8.33
Fe	−2.41	16.6	0.67	7.49	14.7	0.276	7.20
Cu	53.2	59.2	57.0	54.8	61.4	0.840	5.45

Values are the means of 10 dogs per treatment. CD, control diet; AN, control diet supplemented with *Ascophyllum nodosum*; PP, control diet supplemented with *Palmaria palmata*; UP, control diet supplemented with *Undaria pinnatifida*; SJ, control diet supplemented with *Saccharina japonica*. * Significantly different from CD (*p* < 0.05). ° GLM analysis performed after logarithmic transformation to normalize data.

## Data Availability

Data supporting the findings of this study are available from the corresponding author on request.

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
