# Peer review of "Seaweed Supplementation Failed to Affect Fecal Microbiota and Metabolome as Well as Fecal IgA and Apparent Nutrient Digestibility in Adult Dogs"

_animals, 2021, doi:10.3390/ani11082234_

Round 1

Reviewer 1 Report

Line 15- have exhibited

Lines 181-183- The sequence of the primers used should be presented instead of exclusively referring to references.

Line 312 – As the authors acknowledge, the microbiological analysis was not comprehensive enough to draw sound conclusions. Even though in companion animals only feces can be collected for this type of analysis, if the depth of the study had been greater, it could reveal differences not found with the less sensitive and comprehensive methodology used. If no differences were found after detailed and in-depth study, then conclusions of lines 397-399 could be drawn, but not without proper analysis.

The authors should discuss the use of animals of different ages and races, and under different environmental conditions,  and their possible contribution to the apparent lack of differences among groups. Individual variations should be taken into account especially when using small sample groups. Moreover, the Latin square experimental design could have resulted in carry over effects that could potentially have masked differences.

Reviewer 2 Report

General comments

This is an interesting and novel study that evaluated the effects of dietary seaweed supplementation on nutrient digestibility and gut functionality of dogs. The main limitation of the study is that the experimental units were household dogs, which may have contributed to a great variation on data, considering that they presented different ages, body weights, breeds, and is difficult for the authors to have sure about the food management by the owners. It is important that the authors include more information about the dog’s gender, if they are castrated or not, and about the BCS, considering that all these factors may impact the results. The authors might also consider evaluating the variation in each variable between the same dog fed a test diet compared with the same dog fed the control diet, to reduce the effect of individual variations.  

Specific comments

Simple summary

L19: apparent total tract digestibility of nutrients.

Abstract

L27-29: Include the number of replicates per treatment after the replicated Latin square design. Is n=10?

Keywords

L38: Do not repeat words already presented in the title.

Introduction

L46: Move this paragraph after the first sentence. I. ex. …gastrointestinal tract. Among the several…

L80: As intestinal health is difficult to evaluate and to define, I suggest changing here and elsewhere to intestinal functionality.

L84-85: apparent total tract digestibility (ATTD) of nutrients.

Material and Methods

L.94: Please, give more details about the dogs: breeds, how many males and females, castrated or not, and BCS.

L117: Silica was included on top of the food with the seaweed?

L132: Cross over or Latin square design?

L147: EE in acid-hydrolysis?

L161: It is not clear how the total nutrients intake was considered to calculate the ATTD of the diets, considering that you calculate by AIA marker.

L204: Did you tested for data normality? How many replicates per treatment?

Results

Include the dry matter intake.

Table 2:

  • Prefer to use the same name for ammonia: ammonia or NH3.
  • Change VFA to SCFA and BCFA here and elsewhere.
  • Divide the sum of SCFA and the sum of BCFA because they have different physiological roles. I suggest running the statistics for total SCFA and total BCFA.
  • It is interesting to do the sum of biogenic amines too (even if you do not find all of them).

L232: prefer to use “Ca” instead of “this last mineral”.

Table 3:

  • Was the microbial data normally distributed? They usually are not. Please, make it clear in the statistical analysis section. Same for fecal IgA.

Discussion

L264. I suggest moving this paragraph after the previous paragraph (L260-263) and improve the connection between them. I.eg. “However, several literature reviews…”

L274: Do not put one (  ) inside other (   ).

L294: C. perfringens instead of Cl.

L296: Avoid paragraphs with only one sentence.

L297: 10g/kg of diet?

L308: Linseed was the only fiber source used in the CD? The CD did not have prebiotics? I think is important to point it out in the discussion.

L317: may not be 100% representative but it does contain bacteria, endogenous substances, and fermentative end-products of the gut. Most studies on dog and cat nutrition have been investigating fecal components and have been finding many interesting results indicatives of gut functionality.

Round 2

Reviewer 1 Report

Although the authors have not increased the depth of microbiota analysis, which would  more confidently assure that supplementation with algae had no impact on the microbiota, they have nevertheless improved the manuscript.